# Variational Message Passing Neural Network for Maximum-A-Posteriori (MAP) Inference

**Zijun Cui**[1]     **Hanjing Wang**[1]     **Tian Gao**[2]     **Kartik Talamadupula**[2]     **Qiang Ji**[1]

[1]ECSE, Rensselaer Polytechnic Institute
[2]IBM Research

## Abstract

Maximum-A-Posteriori (MAP) inference is a fundamental task in probabilistic inference and belief propagation (BP) is a widely used algorithm for MAP inference. Though BP has been applied successfully to many different fields, it offers no performance guarantee and often performs poorly on loopy graphs. To improve the performance on loopy graphs and to scale up to large graphs, we propose a *variational message passing neural network* (V-MPNN), where we leverage both the power of neural networks in modeling complex functions and the well-established algorithmic theories on variational belief propagation. Instead of relying on a hand-crafted variational assumption, we propose a neural-augmented free energy where a general variational distribution is parameterized through a neural network. A message passing neural network is utilized for the minimization of neural-augmented free energy. Training of the MPNN is thus guided by neural-augmented free energy, without requiring exact MAP configurations as annotations. We empirically demonstrate the effectiveness of the proposed V-MPNN by comparing against both state-of-the-art training-free methods and training-based methods.

## 1 INTRODUCTION

Given a probability distribution of a set of random variables, a Maximum-A-Posteriori (MAP) inference problem involves identifying the most probable configuration of a subset of unobserved random variables with observed evidence for the rest of the variables. MAP inference problem has been studied in different communities, such as discrete energy minimization [Kappes et al., 2013] where optimization solvers are designed to directly solve for the optimal solution (i.e., the most probable configuration). Solving the MAP problem exactly is NP-hard, even with binary variables [Kolmogorov and Zabin, 2004, Cooper, 1990]. MAP inference on a probabilistic graphical model (PGM) is a fundamental task in probabilistic inference, where the joint probability distribution of a set of random variables is captured by a PGM. Such task has lots of real-world applications such as image semantic segmentation in computer vision [Knobelreiter et al., 2020] and protein structure prediction in biochemistry [Soni et al., 2010]. In this work, we focus on MAP inference inside PGM context.

Different probabilistic inference algorithms have been proposed leveraging underlying structures of graphs, with belief propagation (BP) via message passing [Murphy et al., 2013] being a popular and widely used one. Besides, for efficient approximate inference, variational methods have been widely considered whereby probabilistic inference is reformulated as an optimization problem. Variational assumptions are introduced over variational distributions such as mean field assumption [Barabási et al., 1999] and Bethe assumption [Yedidia et al., 2001a]. Under mean field assumption, a variational distribution can be fully factorized which in general does not hold on an arbitrary graph. Bethe assumption is relaxed and is true on loop-free graphs. Variational BP is to perform variational inference through message passing and is theoretically grounded on the well-established connection between BP and Bethe free energy [Tatikonda and Jordan, 2002, Yedidia et al., 2003, 2000, 2001a, Heskes, 2004]. Variational BP under Bethe assumption is exact on loop-free graphs, but its performance on an arbitrary loopy graph remains inaccurate without performance guarantee [Cannings et al., 1976, Shenoy and Shafer, 2008]. Different works based on variational BP have been proposed to improve the performance on loopy graphs, all of which rely on specific variational assumptions, resulting in specific families of variational distributions.

In this work, we propose a *variational message passing neural network* (V-MPNN) for improved MAP inference

*Accepted for the 38th Conference on Uncertainty in Artificial Intelligence* (UAI 2022).

performance on loopy graphs. V-MPNN leverages both the power of neural networks in modeling complex functions and the well-established algorithmic theories on variational BP. In particular, a neural-augmented free energy is proposed where variational distribution is parameterized via a neural network. An optimal variational condition is explored during training. Minimization of neural-augmented free energy is achieved through a message passing neural network (MPNN), which performs probabilistic inference through message passing. The training of the MPNN is guided by neural-augmented free energy, which is different from existing neural-network-based inference methods that require exact inference results as annotations. Without requiring labeled training data, our proposed V-MPNN is data efficient. More importantly, our model can scale up to large graphs where exact inference results are unobtainable.

## 2 RELATED WORKS

**MAP inference.** MAP inference can be directly solved as an integer optimization problem [Wu et al., 2020] or can be relaxed to be a linear optimization problem (LP). With the constraints on marginals enforcing global consistency, i.e., marginal polytope, exact MAP inference can be achieved under LP relaxation [Wainwright and Jordan, 2008]. Marginal polytope is in general intractable. Instead, constraints enforcing local consistency (e.g., pairwise consistency) are considered, that is, local polytope [Sherali and Adams, 1990]. Local polytope yields pseudo-marginals that are local consistent but is not guaranteed to be exact. Unfortunately, MAP inference under LP relaxation with local polytope remains computational prohibitive, particularly on large graphs [Yanover et al., 2006].

**Variational BP for MAP inference.** Variational BP is to perform variational inference through message passing. Variational BP is based on the connection between BP and Bethe free energy [Yedidia et al., 2001b]. Since Bethe free energy can exactly capture only loop-free graphs, BP is guaranteed to be exact on loop-free graphs and is only an approximate inference on loopy graphs. Different techniques have been proposed to improve the performance of BP on loopy graphs, including initialization strategies [Koehler, 2019, Knoll et al., 2018], message update scheduling [Elidan et al., 2012, Knoll et al., 2015, Aksenov et al., 2020] and damping [Murphy et al., 2013, Pretti, 2005]. In addition to these practical techniques, more sophisticated hand-crafted variational distributions are proposed, leading to different variational BP algorithms [Hazan and Shashua, 2010, Riegler et al., 2012]. For example, max-product tree-reweighted message passing (TRW-MP) [Wainwright et al., 2005a] decomposed the original joint distribution into a convex combination of tree-structured distributions. A tree-reweighted variational free energy is correspondingly derived. TRW-MP is guaranteed

to produce exact MAP configurations under a certain condition but it suffers from convergence issues.

Existing studies show that the entropy term within a variational free energy heavily affects the algorithm performance [Ravikumar et al., 2010, Meshi et al., 2012, Lee et al., 2020, Savchynskyy et al., 2011, Hazan and Shashua, 2012]. More specifically, when the entropy is concave and the variational free energy is thus convex, a class of message passing algorithms is obtained with convergence guarantee [Savchynskyy et al., 2011, 2012, Hazan and Shashua, 2012, Weiss et al., 2012, Meshi et al., 2015]. MAP inference error bound with convex free energy can also be derived. In this work, we propose to further reduce the MAP inference error bound by leveraging neural networks.

**Neural networks for probabilistic inference.** Neural networks have been considered for probabilistic inference tasks. Yoon et al. [2019] empirically demonstrated the usage of MPNN [Gilmer et al., 2017] for probabilistic inference, including MAP inference and marginal inference. The architecture of MPNNs follows a message passing scheme. Messages and beliefs are parameterized by neural networks and are learned from observed probabilistic graphs annotated with corresponding exact inference results. Though inspired by belief propagation, MPNN is solely learned from data. Different works have been proposed along this line, the majority of which are for marginal inference. Satorras and Welling [2020] proposed to refine messages from belief propagation via messages learned in MPNN. Kuck et al. [2020] proposed a belief propagation neural network (BPNN) where beliefs are regularized by minimizing a Bethe free energy. Zhang et al. [2019] proposed a factor graph neural network (FGNN) that can perform MAP inference. FGNN is proved to be equivalent to BP and thus can perform well only when ordinary BP does well. Hence, FGNN does not explicitly address the poor inference performance issue of BP on loopy graphs. All the neural-network-based methods mentioned above require either exact MAP configurations or exact partition functions as annotations for fully supervised training. As a result, these methods are limited to small graphs where exact inference results are obtainable.

## 3 PROPOSED METHOD

We propose a variational message passing neural network (V-MPNN) for improving inference performance on loopy graphs and scaling up to large graphs. V-MPNN leverages both the power of neural networks in modeling complex functions and the algorithmic theories on variational BP. We begin with preliminaries that are necessary for later discussions. We then introduce our proposed V-MPNN. Towards the end of this section, we summarize the training objectives of the proposed V-MPNN.

## 3.1 PRELIMINARIES

In this work, we focus on MAP inference on discrete pairwise markov random fields (MRFs). We first define MAP inference on MRFs and then introduce the variational free energy. We discuss different families of variational distributions and introduce the minimization of a variational free energy through message passing. In the end, we show the connection between the optimality of minimizing a variational free energy and the exactness of MAP inference.

### 3.1.1 MAP Inference on Markov Random Field

Given a set of $N$ random variables $\boldsymbol{x} = \{x_1, x_2, ..., x_N\}$ in discrete space $\chi = \chi_1 \times \chi_2 \times ... \times \chi_M$, their joint probability distribution is captured by an MRF $\mathcal{G} = (\mathcal{V}, \mathcal{E})$ where $|\chi_i| = k_i$ is the number of possible states of each variable $x_i$, $|\mathcal{V}| = N$, $|\mathcal{E}| = M$ with $M$ being the total number of edges in the graph. The joint probability distribution of $\boldsymbol{x}$ is defined as,

$$p(\boldsymbol{x}) \propto \exp(\sum_{i \in \mathcal{V}} \theta_i(x_i) + \sum_{(i,j) \in \mathcal{E}} \theta_{ij}(x_i, x_j)) \quad (1)$$

where $\mathcal{E}$ refers to the set $\{(i, j) : i \in \mathcal{V}, j \in \mathcal{N}(i), i < j\}$. $\boldsymbol{\theta}$ defines probability parameters of the graph $\mathcal{G}$. $\theta_i(x_i)$ is the unary potential of variable $x_i$ and $\theta_{ij}(x_i, x_j)$ is the pairwise potential of two neighboring variables $x_i$ and $x_j$ connected via edge $(i, j)$. Given a graph $\mathcal{G}$ and its probability parameters $\boldsymbol{\theta}$, the MAP inference task is formulated as

$$\begin{aligned} \boldsymbol{x}^* &= \arg\max_{\boldsymbol{x} \in \chi} p(\boldsymbol{x}) \\ &= \arg\max_{\boldsymbol{x} \in \chi} \sum_{i \in \mathcal{V}} \theta_i(x_i) + \sum_{(i,j) \in \mathcal{E}} \theta_{ij}(x_i, x_j) \end{aligned} \quad (2)$$

### 3.1.2 Variational Free Energy

Variational method converts a probabilistic inference problem to an optimziation problem, solving for a variational distribution by minimizing a variational free energy [Blei et al., 2017]. Given a target joint distribution $p(\boldsymbol{x})$, Gibbs free energy as a function of a variational distribution $q(\boldsymbol{x})$ is defined as

$$G(q) = U(q) - \mathcal{T}^\circ H(q) \quad (3)$$

$U(q) = \sum_{\boldsymbol{x}} q(\boldsymbol{x}) E(\boldsymbol{x})$ is the average energy and the energy function $E(\boldsymbol{x})$ is specified by $p(\boldsymbol{x})$. $H(q) = -\sum_{\boldsymbol{x}} q(\boldsymbol{x}) \ln q(\boldsymbol{x})$ is the entropy. $\mathcal{T}^\circ$ is the temperature. For MAP inference, temperature is specified to be a sufficiently small value $\epsilon$ ($\mathcal{T}^\circ = \epsilon$). An optimal variational distribution is obtained as

$$q^* = \arg\min_{q \in \mathbb{M}(\mathcal{G})} G(q) \quad (4)$$

Marginal polytope $\mathbb{M}(\mathcal{G})$ enforces global consistency as $\mathbb{M}(\mathcal{G}) = \{q : q \geq 0; \sum_{\boldsymbol{x}} q(\boldsymbol{x}) = 1\}$. This constrained optimization is strictly convex and $q^*$ achieves zero KL divergence w.r.t. the target distribution, that is, $KL(q^*||p) = 0$. Exact inference can be performed with $q^*$. However, minimizing the Gibbs free energy over marginal polytope is in general computational prohibitive. Variational assumptions are introduced for tractable variational distribution.

On pairwise MRF with the joint distribution defined in Eq. 1, we have $E(\boldsymbol{x}) = -\sum_{i \in \mathcal{V}} \theta_i(x_i) - \sum_{(i,j) \in \mathcal{E}} \theta_{ij}(x_i, x_j)$ and the average energy is computed as

$$\begin{aligned} U(q) &= U(\{q_i\}, \{q_{ij}\}) = \\ &-\sum_{i \in \mathcal{V}} \sum_{x_i} q_i(x_i) \theta_i(x_i) - \sum_{(i,j) \in \mathcal{E}} \sum_{x_i, x_j} q_{ij}(x_i, x_j) \theta_{ij}(x_i, x_j) \end{aligned} \quad (5)$$

The average energy becomes a function of local marginals $\{q_i\}_{i \in \mathcal{V}}$ and $\{q_{ij}\}_{(i,j) \in \mathcal{E}}$ with $q_i(x_i) = \sum_{\boldsymbol{x} \setminus x_i} q(\boldsymbol{x})$ and $q_{ij}(x_i, x_j) = \sum_{\boldsymbol{x} \setminus (x_i \cup x_j)} q(\boldsymbol{x})$. We thus assume a variational distribution $q(\boldsymbol{x})$ is a function of $\{q_i(x_i)\}_{i \in \mathcal{V}}$ and $\{q_{ij}(x_i, x_j)\}_{(i,j) \in \mathcal{E}}$, referred to as *pairwise assumption*. Pairwise assumption is widely used on pairwise MRF and there exist various families of variational distributions under pairwise assumption as introduced below.

**Families of variational distributions.** Belief propagation (BP) [Murphy et al., 2013] and TRW-MP [Wainwright et al., 2005a] are the two most representative families of variational distributions under pairwise assumption. In BP, the family of variational distribution is defined as:

$$q^{\text{BP}}(\boldsymbol{x}) = \prod_{i \in \mathcal{V}} q_i(x_i) \prod_{(i,j) \in \mathcal{E}} \frac{q_{ij}(x_i, x_j)}{q_i(x_i) q_j(x_j)} \quad (6)$$

Correspondingly, we obtain a variational free energy (i.e, Bethe free energy):

$$\begin{aligned} &G_{\text{BP}}(\{q_i\}, \{q_{ij}\}) = \\ &U(\{q_i\}, \{q_{ij}\}) - \epsilon(\sum_{i \in \mathcal{V}} (1 - |\mathcal{N}(i)|) H(q_i) + \sum_{(i,j) \in \mathcal{E}} H(q_i, q_j)) \end{aligned} \quad (7)$$

$\mathcal{N}(i)$ denotes the set of neighboring nodes of $i$-th node. $H(q_i) = -\sum_{x_i} q_i(x_i) \ln q_i(x_i)$. $H(q_i, q_j) = -\sum_{x_i, x_j} q_{ij}(x_i, x_j) \ln q_{ij}(x_i, x_j)$. In TRW-MP, a convex combination of tree-structured distributions via spanning trees is employed for approximating probability distribution. The family of variational distribution is defined as

$$q^{\text{TRW-MP}}(\boldsymbol{x}) = \prod_{i \in \mathcal{V}} q_i(x_i) \prod_{(i,j) \in \mathcal{E}} (\frac{q_{ij}(x_i, x_j)}{q_i(x_i) q_j(x_j)})^{\rho_{ij}} \quad (8)$$

which is closely related to BP but differs in terms of an edge appearance probability $\rho_{ij} \in (0, 1]$. Edge appearance probability $\rho_{ij}$ measures the probability of an edge $(i, j)$ in a graph $\mathcal{G}$ being present in a randomly chosen spanning

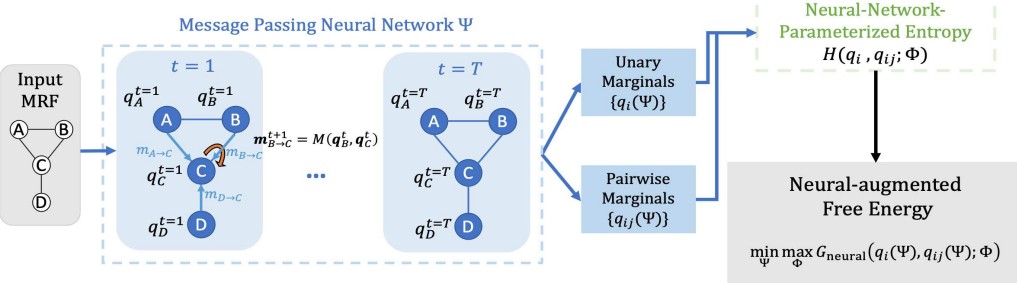

Figure 1: Overview of the proposed variational message passing neural network (V-MPNN)

tree. A variational free energy is correspondingly obtained as

$$G_{\text{TRW-MP}}(\{q_i\}, \{q_{ij}\}) =$$
$$U(\{q_i\}, \{q_{ij}\}) - \epsilon\left(\sum_{i\in\mathcal{V}}(1 - \sum_{j\in\mathcal{N}(i)}\rho_{ij})H(q_i) + \sum_{(i,j)\in\mathcal{E}}\rho_{ij}H(q_i, q_j)\right) \quad (9)$$

TRW-MP is guaranteed to perform exact MAP inference under a certain post-checking condition [Wainwright et al., 2005a,b]. In summary, under the pairwise assumption, a variational free energy is of a general form:

$$G_{\text{pairwise}}(\{q_i\}, \{q_{ij}\}) =$$
$$U(\{q_i\}, \{q_{ij}\}) - \epsilon\left(\sum_{i\in\mathcal{V}}c_i H(q_i) + \sum_{(i,j)\in\mathcal{E}}c_{ij}H(q_i, q_j)\right) \quad (10)$$

Each of the variational BP algorithms (e.g., BP and TRW-MP) is specific to a family of variational distributions, leading to an entropy approximation (i.e., a set of $c_i$ and $c_{ij}$ in Eq. 10). The performance of a variational BP algorithm is hence limited by the corresponding variational assumption. Differently, we propose to leverage the power of a neural network to automatically explore the optimal variational distribution family under the pairwise assumption.

**Minimization of a variational free energy.** Given a variational free energy in Eq. 10, the optimal solution set $\{q_i^*, q_{ij}^*\}_{i\in\mathcal{V},(i,j)\in\mathcal{E}}$ is obtained as:

$$\{q_i^*, q_{ij}^*\} = \arg\min_{\{q_i, q_{ij}\}\in\mathbb{L}(\mathcal{G})} G_{\text{pairwise}}(\{q_i\}, \{q_{ij}\}) \quad (11)$$

with the local polytope constraint set $\mathbb{L}(\mathcal{G}) = \{\{q_i, q_{ij}\} : q_i \geq 0; q_{ij} \geq 0; \sum_{x_i} q_i(x_i) = 1, \forall i \in \mathcal{V}; q_i(x_i) = \sum_{x_j} q_{ij}(x_i, x_j), \forall(i,j) \in \mathcal{E}\}$. This constrained optimization is in general not convex. Its convexity depends on the concavity of the entropy term, which varies with different variational distribution families. Solving for optimal solution can be implemented through message passing. After convergence, fixed-point solutions are guaranteed to be local optimal in minimizing $G_{\text{pairwise}}$. However, a variational gap usually exists between $q^*$ and the target distribution $p$ (i.e., $KL(q^*\|p) > 0$), where $q^*$ is computed from $\{q_i^*, q_{ij}^*\}_{i\in\mathcal{V},(i,j)\in\mathcal{E}}$. MAP inference is performed as

$x_i^* = \arg\max_{x_i} q_i^*(x_i)$. MAP inference is exact if there does not exist a variational gap. Otherwise, the inference remains approximate and is prone to errors.

## 3.2 VARIATIONAL MESSAGE PASSING NEURAL NETWORK

We now introduce the proposed *variational message passing neural network* (V-MPNN). We first introduce the proposed convex neural-augmented free energy whereby we parameterize variational distribution families via a neural network. The proposed neural-augmented free energy is provable convex. The minimal MAP inference error with the proposed neural-augmented free energy is upper bounded by an optimal entropy approximation. We then introduce the minimization of the proposed convex neural-augmented free energy through a message passing neural network (MPNN). The MPNN performs inference through message passing with messages parameterized via neural network parameters. In the end, we summarize the training objectives together with training procedures. The overview of V-MPNN is shown in Figure 1.

### 3.2.1 Convex Neural-augmented Free Energy

Under the pairwise assumption, we introduce the proposed neural-augmented free energy $G_{\text{neural}}$, where we parameterize variational distribution families through neural network parameters $\Phi$. Such parameterization is implicitly achieved via a neural-network-parameterized entropy approximation:

$$G_{\text{neural}}(\boldsymbol{q}^{node}, \boldsymbol{q}^{edge}; \Phi)$$
$$= U(\boldsymbol{q}^{node}, \boldsymbol{q}^{edge}) - \epsilon H(\boldsymbol{q}^{node}, \boldsymbol{q}^{edge}; \Phi) \quad (12)$$

with input tensors $\boldsymbol{q}^{node} = \{q_i\}_{i\in\mathcal{V}} \in \mathbb{R}^{N\times k}$ and $\boldsymbol{q}^{edge} = \{q_{ij}\}_{(i,j)\in\mathcal{E}} \in \mathbb{R}^{M\times k^2}$. The calculation of $U(\boldsymbol{q}^{node}, \boldsymbol{q}^{edge})$ directly follows the definition of the average energy and requires no free parameters to be learned. Neural-network-parameterized entropy approximation is realized through a

neural network with three sets of free parameters $\phi^{node} \in \mathbb{R}^{1 \times N}$, $\phi^{edge} \in \mathbb{R}^{1 \times M}$, $\phi^{\Delta} \in \mathbb{R}^{N \times N}$. In particular, a row-wise entropy calculation w.r.t. each input tensor is firstly performed, producing intermediate values: $\boldsymbol{h}^{node} = \{H(q_i)\}_{i \in \mathcal{V}} \in \mathbb{R}^{N \times 1}$ and $\boldsymbol{h}^{edge} = \{H(q_i, q_j)\}_{(i,j) \in \mathcal{E}} \in \mathbb{R}^{M \times 1}$. The approximate entropy is then computed as

$$
\begin{aligned}
H(\boldsymbol{q}^{node}, \boldsymbol{q}^{edge}; \Phi) = \phi^{node}\boldsymbol{h}^{node} + \\
\exp(\phi^{edge})\boldsymbol{h}^{edge} + \texttt{sum}(\texttt{ReLU}(\phi^{\Delta}) \odot \Delta\boldsymbol{h})
\end{aligned}
\tag{13}
$$

where $\Delta\boldsymbol{h} \in \mathbb{R}^{N \times N}$ with $\Delta h(i,j) = H(q_i, q_j) - H(q_i)$ if $(i,j) \in \mathcal{E}$. Otherwise, $\Delta h(i,j) = 0$. $\odot$ denotes element-wise product. Neural network parameters $\Phi = \{\phi^{node}, \phi^{edge}, \phi^{\Delta}\}$ are unknown and are to be learned. We theoretically prove the convexity of the proposed neural-augmented free energy and the minimal MAP inference error bound through the following propositions.

**Proposition 1.** *Neural-augmented free energy $G_{neural}$ is provable convex with a strictly concave neural-network-parameterized entropy approximation $H(\boldsymbol{q}^{node}, \boldsymbol{q}^{edge}; \Phi)$.*

*Proof:* We prove this proposition by showing the neural-network-parameterized entropy approximation is strictly concave. We first introduce the definition of concave entropy approximation [Heskes, 2004, Weiss et al., 2012]:

**Definition (Concave Entropy Approximation).** An approximate entropy of Eq. 10 is strictly concave over local polytope $\mathbb{L}(\mathcal{G})$ if there exist $\hat{c}_{ij} > 0$, $\hat{\alpha}_{ij} \geq 0$ and $\hat{c}_i$ such that $c_i = \hat{c}_i - \sum_{j \in \mathcal{N}(i)} \hat{\alpha}_{ij}$ and $c_{ij} = \hat{c}_{ij} + \hat{\alpha}_{ij} + \hat{\alpha}_{ji}$. The approximate entropy becomes

$$
\begin{aligned}
H(\{q_i\}, \{q_{ij}\}) = \sum_{i \in \mathcal{V}} \hat{c}_i H(q_i) + \\
\sum_{(i,j) \in \mathcal{E}} \hat{c}_{ij} H(q_i, q_j) + \sum_{i \in \mathcal{V}} \sum_{j \in \mathcal{N}(i)} \hat{\alpha}_{ij}(H(q_i, q_j) - H(q_i))
\end{aligned}
\tag{14}
$$

With any set of parameters $\hat{c}_{ij} > 0$, $\hat{\alpha}_{ij} \geq 0$ and $\hat{c}_i$, the approximate entropy of Eq. 14 is strictly concave. Tensor operation defined in neural-augmented free energy (Eq. 13) is equivalent to Eq. 14, with $\phi^{node}$, $\phi^{edge}$ and $\phi^{\Delta}$ corresponding to $\{\hat{c}_i\}_{i \in \mathcal{V}}$, $\{\hat{c}_{ij}\}_{(i,j) \in \mathcal{E}}$ and $\{\hat{\alpha}_{ij}\}_{i \in \mathcal{V}, j \in \mathcal{N}(i)}$, respectively. $\exp(\cdot)$ ensures the satisfaction of the constraint $\hat{c}_{ij} > 0$. $\texttt{ReLU}(\cdot)$ ensures the satisfaction of the constraint $\hat{\alpha}_{ij} \geq 0$. By definition of concave entropy approximation, the neural-network-parameterized entropy approximation $H(\boldsymbol{q}^{node}, \boldsymbol{q}^{edge}; \Phi)$ is strictly concave. The neural-augmented free energy $G_{neural}$ is thus convex over local polytope $\mathbb{L}(\mathcal{G})$.

We now show the minimal MAP inference error with the proposed neural-augmented free energy is upper bounded by an optimal entropy approximation. We first define the MAP inference error and then present the proposition with its proof. Let $\boldsymbol{q}^*_\Phi$ denote the optimal solution set $\{q^*_{\Phi,i}, q^*_{\Phi,ij}\}_{i \in \mathcal{V}, (i,j) \in \mathcal{E}}$ minimizing the neural-augmented

free energy $G_{neural}$ parameterized by $\Phi$. Given a target probability distribution $p$, the MAP inference error $\Delta_{map}(\boldsymbol{q}^*_\Phi, p)$ is defined as

$$
\begin{aligned}
\Delta_{map}(\boldsymbol{q}^*_\Phi, p) = \sum_{i \in \mathcal{V}} \sum_{x_i}(p_i(x_i) - q^*_{\Phi,i}(x_i))\theta_i(x_i) \\
+ \sum_{(i,j) \in \mathcal{E}} \sum_{x_i, x_j}(p_{ij}(x_i, x_j) - q^*_{\Phi,ij}(x_i, x_j))\theta_{ij}(x_i, x_j)
\end{aligned}
\tag{15}
$$

with $p_i = \sum_{\boldsymbol{x} \backslash x_i} p(\boldsymbol{x})$ and $p_{ij} = \sum_{\boldsymbol{x} \backslash (x_i \cup x_j)} p(\boldsymbol{x})$. By definition, $\Delta_{map}(\boldsymbol{q}^*_\Phi, p) \geq 0$ [Hazan and Shashua, 2010].

**Proposition 2.** *MAP inference error is upper bounded by an entropy approximation scaled by $\epsilon$, i.e.,*

$$
\Delta_{map}(\boldsymbol{q}^*_\Phi, p) \leq \epsilon H(\boldsymbol{q}^*_\Phi; \Phi)
\tag{16}
$$

*The minimal MAP inference error is hence upper bounded by an optimal entropy approximation with $\Phi^* = \arg\min_\Phi H(\boldsymbol{q}^*_\Phi; \Phi)$.*

*Proof:* Given the optimal solution set $\boldsymbol{q}^*_\Phi$ minimizing the neural-augmented free energy $G_{neural}$ parameterized by $\Phi$, we have

$$
G_{neural}(\{q^*_{\Phi,i}\}, \{q^*_{\Phi,ij}\}; \Phi) \leq G_{neural}(\{p_i\}, \{p_{ij}\}; \Phi) \tag{17}
$$

By reorganizing the above equation, we have

$$
\Delta_{map}(\boldsymbol{q}^*_\Phi, p) \leq \epsilon(H(\boldsymbol{q}^*_\Phi; \Phi) - H(\{p_i\}, \{p_{ij}\}; \Phi)) \tag{18}
$$

Given the fact that $H(\{p_i\}, \{p_{ij}\}) \geq 0$, we can have

$$
\Delta_{map}(\boldsymbol{q}^*_\Phi, p) \leq \epsilon H(\boldsymbol{q}^*_\Phi; \Phi) \tag{19}
$$

With an optimal set of neural network parameters $\Phi^* = \arg\min_\Phi H(\boldsymbol{q}^*_\Phi; \Phi)$, the error bound becomes

$$
\Delta_{map}(\boldsymbol{q}^*_{\Phi^*}, p) \leq \epsilon H(\boldsymbol{q}^*_{\Phi^*}; \Phi^*) \tag{20}
$$

We thus show that the minimal MAP inference error is upper bounded by an optimal entropy approximation. Proposition 2 is the basis for the proposed V-MPNN and it motivates the training of our neural-network-parameterized entropy approximation as we will introduce in Section 3.2.3. In the end, we provide a brief comparison between the proposed neural-augmented free energy and existing variational BP algorithms:

**Proposition 3.** *Neural-augmented free energy subsumes existing variational distribution families (e.g., BP and TRW-MP) as a strict generalization. The optimal MAP inference performance achieved with neural-augmented free energy is superior or comparable to existing variational distribution families, i.e., $\Delta_{map}(\boldsymbol{q}^*_{\Phi^*}, p) \leq \Delta_{map}(\boldsymbol{q}^*_{\Phi_{fix}}, p)$*

*Proof:* By manipulating neural network parameters, different existing variational distribution families can be realized

with neural-augmented free energy. For example, neural-augmented free energy with $\Phi$ specified as $\phi^{node} = 1 - |\mathcal{N}(i)|$, $\phi^{edge} = 1$ and $\phi^{\Delta} = 0$ is equivalent to BP. Furthermore, given the fact that $\boldsymbol{q}^*_{\Phi^*} = \arg\min_{\boldsymbol{q}} G_{\text{neural}}(\boldsymbol{q}; \Phi^*)$, we have

$$U(\boldsymbol{q}^*_{\Phi^*}) - \epsilon H(\boldsymbol{q}^*_{\Phi^*}; \Phi^*) \leq U(\boldsymbol{q}^*_{\Phi^{fix}}) - \epsilon H(\boldsymbol{q}^*_{\Phi^{fix}}; \Phi^*) \quad (21)$$

with $\boldsymbol{q}^*_{\Phi^{fix}}$ denotes the optimal variational distribution minimizing neural-augmented free energy specified with fixed parameters $\Phi^{fix}$. By subtracting $U(\{p_i\}, \{p_{ij}\})$ on both sides of Eq. 21 and a re-organization, we have

$$\Delta_{map}(\boldsymbol{q}^*_{\Phi^*}, p) \leq \Delta_{map}(\boldsymbol{q}^*_{\Phi^{fix}}, p) + \epsilon\Delta \quad (22)$$

with $\Delta = H(\boldsymbol{q}^*_{\Phi^*}; \Phi^*) - H(\boldsymbol{q}^*_{\Phi^{fix}}; \Phi^*)$. If $\Delta \leq 0$, it is clear that $\Delta_{map}(\boldsymbol{q}^*_{\Phi^*}, p) \leq \Delta_{map}(\boldsymbol{q}^*_{\Phi^{fix}}, p)$. If $\Delta > 0$, we can have $\Delta_{map}(\boldsymbol{q}^*_{\Phi^*}, p) \leq \Delta_{map}(\boldsymbol{q}^*_{\Phi^{fix}}, p)$ with a sufficiently small coefficient ($\epsilon \to 0$). To show the latter point, we first note that $\Delta = H(\boldsymbol{q}^*_{\Phi^*}; \Phi^*) - H(\boldsymbol{q}^*_{\Phi^{fix}}; \Phi^*) \leq H(\boldsymbol{q}^*_{\Phi^*}; \Phi^*)$. We then show that $H(\boldsymbol{q}^*_{\Phi^*}; \Phi^*)$ is upper bounded by a constant and finite value $\delta$. For clear derivation, we use notation $\boldsymbol{q}$ and $\Phi$ and derive for $H(\boldsymbol{q}; \Phi)$ in the following. The derivation applies to arbitrary $\boldsymbol{q}$ and $\Phi$. To derive $\delta$, we firstly re-organize the entropy approximation $H(\boldsymbol{q}; \Phi)$ defined in Eq. 13 as

$$H(\boldsymbol{q}; \Phi) = \sum_{i \in \mathcal{V}} (\phi_i^{node} - \sum_{j \in \mathcal{N}(i)} \text{ReLU}(\phi_{ij}^{\Delta}))H(q_i) +$$
$$\sum_{(i,j) \in \mathcal{E}} (\exp(\phi_{ij}^{edge}) + \text{ReLU}(\phi_{ij}^{\Delta}) + \text{ReLU}(\phi_{ji}^{\Delta}))H(q_i, q_j) \quad (23)$$

We now show that both $H(q_i)$ and $H(q_i, q_j)$ are bounded by a constant value. For $H(q_i)$, applying the Jensen's inequality yields,

$$H(q_i) = -\sum_{x_i} q_i(x_i) \log q_i(x_i) = \sum_{x_i} q_i(x_i) \log \frac{1}{q_i(x_i)}$$
$$\leq \log \sum_{x_i} \frac{q_i(x_i)}{q_i(x_i)} = \log k_i \quad (24)$$

where $k_i$ indicates the number of states of variable $x_i$. Similarly, we have $H(q_i, q_j) \leq \log k_{ij}$ where $k_{ij}$ indicates the number of joint configurations of variables $x_i$ and $x_j$. Given the bounds for $H(q_i)$ and $H(q_i, q_j)$, we can conclude

$$H(\boldsymbol{q}; \Phi) \leq \delta = \sum_{i \in \mathcal{V}} |\phi_i^{node} - \sum_{j \in \mathcal{N}(i)} \text{ReLU}(\phi_{ij}^{\Delta})| \log k_i +$$
$$\sum_{(i,j) \in \mathcal{E}} (\exp(\phi_{ij}^{edge}) + \text{ReLU}(\phi_{ij}^{\Delta}) + \text{ReLU}(\phi_{ji}^{\Delta})) \log k_{ij} \quad (25)$$

$\delta$ is only a function of underlying graph and parameters $\Phi$. Thus, we have $\Delta \leq H(\boldsymbol{q}^*_{\Phi^*}; \Phi^*) \leq \delta$. With $\Delta \leq \delta$, we now can show

$$\Delta_{map}(\boldsymbol{q}^*_{\Phi^*}, p) \leq \Delta_{map}(\boldsymbol{q}^*_{\Phi^{fix}}, p) + \epsilon\delta \quad (26)$$

$\delta$ is not a function of $\epsilon$. Furthermore, given the mild assumption that neural parameters $\Phi$ are of finite values, $\delta$

is always finite. With a constant and finite upper bound $\delta$, there always exists a sufficiently small $\epsilon$ such that $\Delta_{map}(\boldsymbol{q}^*_{\Phi^*}, p) \leq \Delta_{map}(\boldsymbol{q}^*_{\Phi^{fix}}, p)$. Theoretically, we show that the optimal MAP inference performance achieved with neural-augmented free energy is superior or comparable to existing variational distribution families.

### 3.2.2 Minimization of Neural-augmented Free Energy with MPNN

To minimize the neural-augmented free energy, we employ a message passing neural network (MPNN). In particular, $\boldsymbol{q}^{node} = \{q_i\}_{i \in \mathcal{V}}$ and $\boldsymbol{q}^{edge} = \{q_{ij}\}_{(i,j) \in \mathcal{E}}$ are parameterized by an MPNN, leading to unary marginal estimate $\boldsymbol{q}^{node}(\Psi)$ and pairwise marginal estimate $\boldsymbol{q}^{edge}(\Psi)$. We detail the MPNN module in the following.

**Unary marginal estimation.** We map each node in MPNN to a variable in MRF with hidden feature $\boldsymbol{h}_i \in R^{k_i}$. $k_i$ is the number of possible states of variable $x_i$. In total, we have node features $\boldsymbol{h} = \{\boldsymbol{h}_1, \boldsymbol{h}_2, ..., \boldsymbol{h}_N\}$ and $N$ is the total number of nodes. Node feature $\boldsymbol{h}_i$ corresponds to the unary marginal estimation in logarithmic space, up to a scale factor $z_i$. At every iteration $t$, each node receives a message from each of its neighboring nodes as

$$\boldsymbol{m}_{j \to i}^{t+1} = \mathcal{M}(\boldsymbol{h}_i^t, \boldsymbol{m}_{i \to j}^t, \theta_{ij}, z_j^t) \quad (27)$$

$\mathcal{M}$ is a message function realized via a multi-layer perceptron (MLP). The messages are then aggregated through summation, i.e., $\boldsymbol{m}_i^{t+1} = \sum_{j \in \mathcal{N}(i)} \boldsymbol{m}_{j \to i}^{t+1}$. Each node then updates its hidden state with the aggregated message as:

$$\boldsymbol{h}_i^{t+1} = \mathcal{U}(\boldsymbol{m}_i^{t+1}, \theta_i, z_i) = \boldsymbol{m}_i^{t+1} + \theta_i - \ln(z_i^{t+1}) \quad (28)$$

$\mathcal{U}$ is a node update function and is customized based on BP's belief equation, instead of employing a gated recurrent unit (GRU) as a standard MPNN. Scale factor $z_i$ is calculated as $z_i^{t+1} = \sum_{x_i} \exp(\theta_i(x_i) + \boldsymbol{m}_i^{t+1}(x_i))$. The update process is repeated until convergence. Estimated marginal probability of variable $x_i$ (i.e., $q_i$) is obtained as

$$q_i = \exp(\boldsymbol{h}_i^{(T)}) \quad (29)$$

where $\boldsymbol{h}_i^{(T)}$ is the hidden feature from the last iteration.

**Pairwise marginal estimation.** The pairwise marginal estimation is obtained as

$$q_{ij} = \exp(\theta_{ij} + \boldsymbol{h}_i^{(T)} + \boldsymbol{h}_j^{(T)} - \boldsymbol{m}_{j \to i}^T - \boldsymbol{m}_{i \to j}^T) \quad (30)$$

where $\boldsymbol{h}_i^{(T)}$ and $\boldsymbol{h}_j^{(T)}$ are the respective hidden features for $i$-th node and $j$-th node from the last iteration of MPNN. Eq. 30 is defined based on BP's pairwise belief equation.

We customize our MPNN based on BP with only message function $\mathcal{M}$ containing free parameters that need to be learned. The free parameters of MPNN $\Psi$ hence refers to parameters within the message function $\mathcal{M}$.

### 3.2.3 Training Objectives

In summary, we have two sets of parameters to be learned: $\Phi$ and $\Psi$. The total training objective is based on neural-augmented free energy, i.e.,

$$\min_{\Psi} \max_{\Phi} G_{\text{neural}}(\boldsymbol{q}^{node}(\Psi), \boldsymbol{q}^{edge}(\Psi); \Phi) \qquad (31)$$

under the local polytope constraint $\mathbb{L}(\mathcal{G})$. To effectively perform the training with the neural-augmented free energy, we consider a two-phase alternative update. For each iteration $r$, we first update $\Psi$ given the neural-augmented free energy specified with current $\Phi^r$, i.e.,

$$\Psi^{r+1} = \arg \min_{\Psi} G_{\text{neural}}(\boldsymbol{q}^{node}(\Psi), \boldsymbol{q}^{edge}(\Psi); \Phi^r) \quad (32)$$

The constraints within local polytope $\mathbb{L}(\mathcal{G})$ are naturally satisfied by adopting BP belief equations for customizing our MPNN. We then update $\Phi$. By definition of $G_{\text{neural}}$ in Eq. 12, we have $\max_{\Phi} G_{\text{neural}}(\Phi) = \min_{\Phi} H(\Phi)$ and the $\Phi$ is updated as

$$\Phi^{r+1} = \arg \min_{\Phi} H(\boldsymbol{q}^{node}(\Psi^{r+1}), \boldsymbol{q}^{edge}(\Psi^{r+1}); \Phi) \quad (33)$$

Following proposition 2, we theoretically prove that the entropy is the upper bound of the MAP inference error and hence updating $\Phi$ by minimizing the entropy is equivalent to minimizing the MAP inference error. We update two sets of parameters alternatively until convergence. After training, only the MPNN module with the optimal parameters $\Psi^*$ is required for MAP inference. MAP configuration is obtained as

$$x_i^* = \arg \max_{x_i \in \chi_i} q_i(x_i; \Psi^*) \qquad (34)$$

## 4 EXPERIMENTS

**Datasets.** We consider 13 classic graphs for evaluation – these are the most representative graphs of real world models, and are employed widely in related works [Yoon et al., 2019]. Their structures are illustrated in Figure 2. There are three loop-free graphs, i.e., STAR, TREE and PATH. The other 10 graphs are loopy graphs, with the COMPLETE graph being the most complex one. To simulate graphical models with different parameters, we randomly sample from uniform distributions [Wainwright et al., 2005a]. Particularly, we assume $\theta_i(x_i) = b_i x_i$ and $\theta_{ij}(x_i, x_j) = J_{ij} x_i x_j$ with $x_i = \{-1, 1\}$. Pairwise parameters $J_{ij}$ are sampled from a uniform distribution, i.e., $J_{ij} = J_{ji} \sim U[-1, 1]$. Unary parameters $b_i$ are sampled from a uniform distribution as $b_i \sim U[-0.05, 0.05]$. For each type of graph, we simulate 1000 graphs for training and 100 graphs for testing. GT MAP configuration of each simulated graph is computed by enumeration. Since enumeration is a computationally expensive process, we limit the sizes of the graphs. Particularly, we consider two graph sizes: N=9 and N=15.

**Evaluation metrics.** We employ the accuracy of estimated MAP configuration as the evaluation metric [Yoon et al., 2019]. Given a GT MAP configuration $\boldsymbol{x}^* = \{x_1^*, ..., x_N^*\}$, and an estimated MAP configuration $\hat{\boldsymbol{x}} = \{\hat{x}_1, ..., \hat{x}_N\}$, the accuracy of $\hat{\boldsymbol{x}}$ is calculated as $\frac{\#(x_i^* = \hat{x}_i)}{N}$. We report the average accuracy over testing graphs.

**Experiment settings.** ADAM optimizer is employed for training with a learning rate $1e - 4$. In Eq 12, $\epsilon = 0.0001$. For the message function $\mathcal{M}$, a five-layer MLP is adopted and hidden features are of dimension 256. Messages propagate for $T = 10$ iterations. MPNN is pre-trained at a message level, where a mean squared error between messages from $\mathcal{M}$ and messages from BP is used as the loss function.

### 4.1 COMPARISON TO STATE-OF-THE-ART METHODS

We compare the proposed V-MPNN to different state-of-the-art methods for approximate MAP inference. Specifically, we consider both training-free methods and training-based methods for comparison. Training-free methods refer to optimization algorithms that do not contain neural network components and thus require no training procedure, such as the belief propagation algorithm. In this work, we limit our comparisons to message-passing-based optimization approaches. Training-based methods refer to neural-network-based methods for probabilistic inference tasks.

#### 4.1.1 Comparison to Training-free Methods

We consider three training-free methods: BP [Murphy et al., 2013], TRW-MP [Wainwright et al., 2005a] and max product linear programming (MPLP) [Globerson and Jaakkola, 2007]. For all these three methods, we apply the same stopping criterion: if the maximum number of iteration $t$ is larger than 200 or the average difference between beliefs from two consecutive iterations is sufficiently small, i.e., $\frac{1}{N}\sum_{i=1}^{N}|b_i^{t+1} - b_i^t|^2 < 1e - 7$, we break the algorithm and obtain the estimated inference results[1]. Following [Wainwright et al., 2005a], for both BP and TRW-MP, we apply message damping in log-space with damping parameter set to be 0.5. The edge appearance probability in TRW-MP is set as $\rho_{ij} = \frac{|\mathcal{V}| - 1}{|\mathcal{E}|}$.

Results are presented in Table 1. As shown, we can see that V-MPNN achieves the best average accuracy with both sizes of graphs. On each type of graph, V-MPNN achieves overall better performance than the other three baselines. On loopy graphs, though the performance of all the algorithms decreases as the complexity of the graph increases, V-MPNN achieves better accuracy compared to the other three baselines. On CIRCULAR LADDER with 15

---

[1]The maximum number of iterations is set to be 200 because the number of converging runs stops changing after 200.

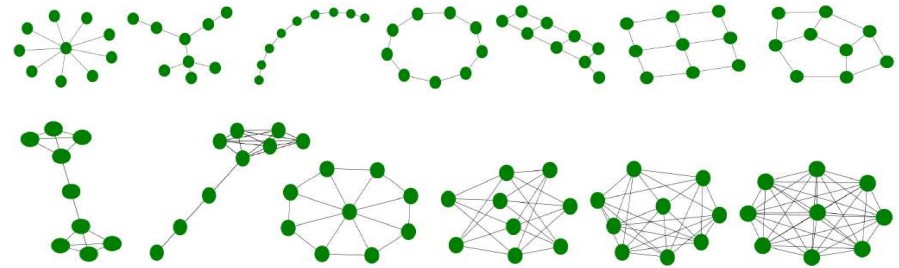

Figure 2: Structures of 13 classic graphs with 9 nodes. Graphs on the first row from left to right are: STAR, TREE, PATH, CIRCLE, LADDER, 2D GRID, CIRCULAR LADDER; graphs on the second row from left to right are: BARBELL, LOLLIPOP, WHEEL, BIPARTITE, TRIPARTITE, COMPLETE

Table 1: Comparison to training-free methods

| Graph | N=9 | | | | N=15 | | | |
|---|---|---|---|---|---|---|---|---|
| | BP | TRW-BP | MPLP | V-MPNN | BP | TRW-BP | MPLP | V-MPNN |
| STAR | **1.0** | .99 | **1.0** | .93 | **1.0** | **1.0** | **1.0** | .74 |
| TREE | **1.0** | .99 | **1.0** | .96 | **1.0** | **1.0** | **1.0** | .93 |
| PATH | **1.0** | **1.0** | **1.0** | .97 | **1.0** | **1.0** | **1.0** | .93 |
| CYCLE | **.91** | .76 | .90 | .85 | .84 | .84 | **.89** | .87 |
| LADDER | .68 | .66 | .72 | **.77** | .63 | .61 | .67 | **.72** |
| 2D GRID | .57 | .48 | **.74** | **.74** | .56 | .50 | .63 | **.69** |
| CIRCULAR LADDER | .62 | .50 | .76 | **.83** | .61 | .53 | .63 | **.73** |
| BARBELL | .57 | .55 | .67 | **.71** | .60 | .57 | .64 | **.66** |
| LOLLIPOP | .59 | .60 | .61 | **.88** | .62 | .55 | .58 | **.67** |
| WHEEL | .56 | .44 | .62 | **.70** | .58 | .50 | .62 | **.69** |
| BIPARTITE | .54 | .52 | .62 | **.74** | .62 | .56 | .55 | **.64** |
| TRIPARTITE | .57 | .62 | .52 | **.68** | .52 | .55 | .51 | **.65** |
| COMPLETE | .56 | .60 | .49 | **.65** | .54 | .54 | .53 | **.60** |
| **MEAN** | .71 | .67 | .73 | **.80** | .70 | .67 | .69 | **.73** |

nodes, V-MPNN achieves 73% accuracy, which is 20% higher than the accuracy achieved by TRW-BP. On loop-free graphs, such as STAR, TREE, and PATH, BP is guaranteed to produce the exact MAP configuration, and thus always achieves 100% accuracy. Though the proposed V-MPNN is theoretically shown to be a strict generalization of BP, training of the MPNN is not guaranteed to find the global optimal, leading to MAP inference errors on loop-free graphs.

### 4.1.2 Comparison to Training-based Methods

We compare the proposed V-MPNN to a training-based method: node-GNN [Yoon et al., 2019] for MAP inference. Node-GNN[2] is the state-of-the-art method that employs neural networks for probabilistic inference tasks. We employ the suggested hyper-parameter settings stated in the paper to perform the experiments.

Results are presented in Table 2. C-LADDER denotes CIR-CULAR LADDER. As shown, V-MPNN achieves significant

---
[2]https://github.com/ks-korovina/pgm_graph_inference.

Table 2: Comparison to training-based method

| Graph | N=9 | | N=15 | |
|---|---|---|---|---|
| | Node-GNN | V-MPNN | Node-GNN | V-MPNN |
| STAR | .65 | **.93** | .52 | **.74** |
| TREE | .77 | **.96** | .75 | **.93** |
| PATH | .81 | **.97** | .73 | **.93** |
| CYCLE | .79 | **.85** | .75 | **.87** |
| LADDER | .72 | **.77** | .69 | **.72** |
| 2D GRID | .72 | **.74** | .74 | .69 |
| C-LADDER | .81 | **.83** | .71 | **.73** |
| BARBELL | **.72** | .71 | .71 | .66 |
| LOLLIPOP | .72 | **.88** | .69 | .67 |
| WHEEL | .68 | **.70** | .70 | .69 |
| BIPARTITE | **.75** | .74 | .74 | .64 |
| TRIPARTITE | **.73** | .68 | .72 | .65 |
| COMPLETE | **.82** | .65 | .70 | .60 |
| **MEAN** | .75 | **.80** | .70 | **.73** |

better average accuracy with both sizes of graphs without requiring exact MAP configurations for training. Across different types of graphs, V-MPNN achieves overall better performance than node-GNN and significantly outperforms node-GNN on loop-free graphs. On TREE with 9 nodes, V-MPNN achieves 96% accuracy, which is 19% higher

than the accuracy achieved by node-GNN. These results show that, under the guidance of well-established algorithmic knowledge, the proposed V-MPNN can be trained to achieve outstanding performance, without requiring exact MAP configurations as annotations.

## 4.2 ABLATION STUDY

In our experiments, MPNN is pre-trained with BP's message equation. In practice, we find this pre-training step is crucial since directly using the neural-augmented free energy objective $G_{\text{neural}}$ without pre-training can easily make training diverge. We thus adopt the pre-training step and then fine tune the model with $G_{\text{neural}}$. To better understand the effectiveness of the neural-augmented free energy objective $G_{\text{neural}}$, we perform an ablation study. Particularly, we compare the performance of V-MPNN to the performance of V-MPNN with pre-training only. We consider 13 classic graphs with 9 nodes. Results are shown in Table 3.

Table 3: Effectiveness of NFE update (N=9)

| Graph | pre-training | pre-training+fine tuning |
|---|---|---|
| STAR | **.93** | **.93** |
| TREE | **.96** | **.96** |
| PATH | **.97** | **.97** |
| CYCLE | .80 | **.85** |
| LADDER | **.77** | **.77** |
| 2D GRID | .72 | **.74** |
| C-LADDER | .82 | **.83** |
| BARBELL | .70 | **.71** |
| LOLLIPOP | **.88** | **.88** |
| WHEEL | **.70** | **.70** |
| BIPARTITE | .72 | **.74** |
| TRIPARTITE | .66 | **.68** |
| COMPLETE | .64 | **.65** |
| **MEAN** | **.79** | **.80** |

As shown, fine-tuning through $G_{\text{neural}}$ improves V-MPNN's average performance compared to V-MPNN with pre-training only. On CYCLE, the performance of V-MPNN is improved by $5\%$ with $G_{\text{neural}}$ through fine tuning. From the results, we can see that the neural-augmented free energy $G_{\text{neural}}$ introduces important effect on the inference performance of V-MPNN, particularly on loopy graphs.

## 5 CONCLUSION

In this work, we proposed a variational message passing neural network for MAP inference. Instead of relying on a specific family of variational distributions, we proposed a neural-augmented free energy where variational assumptions are parameterized via a neural network. An optimal family of variational distributions is learned through training. An MPNN is employed for efficient inference through message passing. Training of the MPNN is performed under the guidance of neural-augmented free energy, without requiring exact MAP configurations as annotations. In our experiments, the proposed V-MPNN outperforms both state-of-the-art training-free and training-based methods for MAP inference, demonstrating the effectiveness of the proposed method.

## ACKNOWLEDGMENTS

This work is supported by the Rensselaer-IBM AI Research Collaboration (http://airc.rpi.edu), part of the IBM AI Horizons Network (http://ibm.biz/AIHorizons).

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
