# OpenReview forum: "Variational Message Passing Neural Network for Maximum-A-Posteriori Inference"
_auai.org/UAI/2022/Conference — UAI 2022 Poster_

### Official Review · Reviewer_P8Gi · 2022-04-08

**Q2(1) Originality/Novelty:** 3
**Q2(2) Significance/Impact:** 3
**Q2(3) Correctness/Technical Quality:** 4
**Q2(6) Clarity Of Writing:** 3
**Q6 Overall Score:** 7
**Q8 Confidence In Your Score:** 4

**Q1 Summary And Contributions:**

The author proposes a variational message passing neural network (V-MPNN) for improved MAP inference performance on loopy graphs.


**Q2 Assessment Of The Paper:**

More detailed information regarding each of these aspects is given below:

**Q2(4) Quality Of Experiments (Optional):**

3: Good: The experimental evaluation is adequate, and the results convincingly support the main claims.

**Q2(5) Reproducibility:**

3: Good: Key resources (e.g., proofs, code, data) are available and key details (e.g., proofs, experimental setup) are sufficiently well-described for competent researchers to confidently reproduce the main results.

**Q3 Main Strengths:**

The author proposes a variational message passing neural network (V-MPNN) for improved MAP inference performance on loopy graphs.
-	A neural free energy is proposed where variational distribution is parameterized via a neural network.  The structure is novel.
-       Proposed a reasonable convex neural free energy function with a proper concave entropy approximation.
-	Minimization of neural free energy is achieved through message passing neural networks (MPNNs).
-       The upper bound of MAP inference error is provided.
-	The training of V-MPNN doesn’t require exact inference results as annotations, i.e. it doesn’t require labeled training data.
-	The model can scale up to large graphs where exact inference results are unobtainable. V-MPNN outperforms state-of-the-art methods (especially in loopy cases) without requiring exact MAP configurations for training.



**Q4 Main Weakness:**

The quality of the MAP inference result relies on neural-network-parameterized strictly concave entropy approximation, which can be a limitation of their work.

**Q5 Detailed Comments To The Authors:**

“Inference in Ising models by graph neural networks with structural features” by HT Huynh could be an interesting comparison in loop-free cases (perhaps loopy cases as well).

“Non-approximate Inference for Collective Graphical Models on Path Graphs via Discrete Difference of Convex Algorithm” by Yasunori Akagi proposed a non-approximate method. Maybe the authors can compare with their work.


**Q7 Justification For Your Score:**

- The proposed neural free energy function and the corresponding graph network structure are novel.
- The paper clearly described the motivation and method.
- Sufficient experimental support.
- No obvious flaws in the proofs.


**Q9 Complying With Reviewing Instructions:**

1: Yes.

---

### Official Review · Reviewer_dsri · 2022-04-11

**Q2(1) Originality/Novelty:** 3
**Q2(2) Significance/Impact:** 3
**Q2(3) Correctness/Technical Quality:** 3
**Q2(6) Clarity Of Writing:** 4
**Q6 Overall Score:** 7
**Q8 Confidence In Your Score:** 2

**Q1 Summary And Contributions:**

MAP inference is a crucial tool for Bayesian learning in general. The authors proposed a framework called V-MPNN to solve the task. The novelty of this paper is the authors proposed a neural energy heuristic that provides some theoretical justifications.

**Q2 Assessment Of The Paper:**

More detailed information regarding each of these aspects is given below:

**Q2(4) Quality Of Experiments (Optional):**

3: Good: The experimental evaluation is adequate, and the results convincingly support the main claims.

**Q2(5) Reproducibility:**

3: Good: Key resources (e.g., proofs, code, data) are available and key details (e.g., proofs, experimental setup) are sufficiently well-described for competent researchers to confidently reproduce the main results.

**Q3 Main Strengths:**

The study is comprehensive enough with theoretical justification and practical verifications.

**Q4 Main Weakness:**

I am not very familiar with the topic, so I cannot provide any sure criticism of the proposed methods.

**Q5 Detailed Comments To The Authors:**

Regardless of the probabilistic graphic model context, can the authors provide some practical use cases for the proposed methods? As far as I know, MAP is often used as the downstream tool for Bayesian learning. Can the authors provide some practical experiments?

**Q7 Justification For Your Score:**

I am not very familiar with the topic of interest, so my scores are based on the paper's presentation only.

**Q9 Complying With Reviewing Instructions:**

1: Yes.

---

### Official Review · Reviewer_SD1U · 2022-04-12

**Q2(1) Originality/Novelty:** 3
**Q2(2) Significance/Impact:** 3
**Q2(3) Correctness/Technical Quality:** 2
**Q2(6) Clarity Of Writing:** 3
**Q6 Overall Score:** 5
**Q8 Confidence In Your Score:** 3

**Q1 Summary And Contributions:**

See detailed comments.

**Q2 Assessment Of The Paper:**

More detailed information regarding each of these aspects is given below:

**Q2(4) Quality Of Experiments (Optional):**

2: Fair: The experimental evaluation is weak: important baselines are missing, or the results do not adequately support the main claims.

**Q2(5) Reproducibility:**

2: Fair: Key resources (e.g., proofs, code, data) are unavailable but key details (e.g., proof sketches, experimental setup) are sufficiently well-described for an expert to confidently reproduce the main results.

**Q3 Main Strengths:**

See detailed comments.

**Q4 Main Weakness:**

See detailed comments.

**Q5 Detailed Comments To The Authors:**

The authors introduce "neural free energy" to perform MAP inference in MRF. This parametrised form of entropy attempts to generalise previously considered variational families.

I find this paper interesting to read. As opposed to the standard approach of proposing extended variational family, the authors propose to alter only the entropy function by introducing additional free variables. To be honest, I find the name neural free energy an over kill, given the simplicity of functional form of free variables.

Overall, the paper is well-written and clear. The approach is novel to the best of my knowledge.

As for corectness, I have questions regarding Prepositions 2/3. First, in Preposition 2 the bound is not known a prior and the neural energy is not bounded. What are the practical benefits/implications of this proposition?
In the proof of Preposition 3 you say "If ∆ > 0,we can have ∆ (q∗∗,p) ≤ ∆ (q∗ ,p) with a sufficiently map Φ map Φfix small coefficient (ε → 0)". However, the optimal variational posterior depends on epsilon, hence delta changes when epsilon changes. Could you clarify the last step of the proof?
How do you want to pick sufficiently small epsilon when delta changes?

The experiments demonstrate the method achieves better empirical performance compared to the baselines.

Overall I recommend borderline accept. To improve or maintain the score I'd need to hear authors' response regarding Prepositions 2/3.

**Q7 Justification For Your Score:**

See detailed comments.

**Q9 Complying With Reviewing Instructions:**

1: Yes.

---

### Decision · Program_Chairs · 2022-05-15

**Decision:**

Accept (Poster)

**Comment:**

Meta Review: The paper proposes a learned variational approximation and corresponding Message Passing Neural Network.  The reviewers found the idea of a trainable variational approach interesting, as well as the fact that no MAP assignments are needed for training.